# Korean Red Ginseng Improves Astrocytic Mitochondrial Function by Upregulating HO-1-Mediated AMPKα–PGC-1α–ERRα Circuit after Traumatic Brain Injury

**DOI:** 10.3390/ijms222313081

**Published:** 2021-12-03

**Authors:** Minsu Kim, Joohwan Kim, Sunhong Moon, Bo Young Choi, Sueun Kim, Hui Su Jeon, Sang Won Suh, Young-Myeong Kim, Yoon Kyung Choi

**Affiliations:** 1Bio/Molecular Informatics Center, Department of Bioscience and Biotechnology, Konkuk University, Seoul 05029, Korea; kjmmmmm6@konkuk.ac.kr (M.K.); sunhong95047@konkuk.ac.kr (S.M.); rlatndms0325@konkuk.ac.kr (S.K.); heeth313@konkuk.ac.kr (H.S.J.); 2Department of Molecular and Cellular Biochemistry, School of Medicine, Kangwon National University, Chuncheon 24341, Gangwon-do, Korea; joohwank@hs.uci.edu (J.K.); ymkim@kangwon.ac.kr (Y.-M.K.); 3Department of Physiology, College of Medicine, Hallym University, Chuncheon 24252, Gangwon-do, Korea; bychoi@hallym.ac.kr (B.Y.C.); swsuh@hallym.ac.kr (S.W.S.)

**Keywords:** Korean red ginseng, heme oxygenase-1, traumatic brain injury, AMPKα–PGC-1α–ERRα circuit, astrocytic mitochondrial function

## Abstract

Heme oxygenase-1 (HO-1) exerts beneficial effects, including angiogenesis and energy metabolism via the peroxisome proliferator-activating receptor-γ coactivator-1α (PGC-1α)–estrogen-related receptor α (ERRα) pathway in astrocytes. However, the role of Korean red ginseng extract (KRGE) in HO-1-mediated mitochondrial function in traumatic brain injury (TBI) is not well-elucidated. We found that HO-1 was upregulated in astrocytes located in peri-injured brain regions after a TBI, following exposure to KRGE. Experiments with pharmacological inhibitors and target-specific siRNAs revealed that HO-1 levels highly correlated with increased AMP-activated protein kinase α (AMPKα) activation, which led to the PGC-1α-ERRα axis-induced increases in mitochondrial functions (detected based on expression of cytochrome c oxidase subunit 2 (MTCO2) and cytochrome c as well as O_2_ consumption and ATP production). Knockdown of ERRα significantly reduced the p-AMPKα/AMPKα ratio and PGC-1α expression, leading to AMPKα–PGC-1α–ERRα circuit formation. Inactivation of HO by injecting the HO inhibitor Sn(IV) protoporphyrin IX dichloride diminished the expression of p-AMPKα, PGC-1α, ERRα, MTCO2, and cytochrome c in the KRGE-administered peri-injured region of a brain subjected to TBI. These data suggest that KRGE enhanced astrocytic mitochondrial function via a HO-1-mediated AMPKα–PGC-1α–ERRα circuit and consequent oxidative phosphorylation, O_2_ consumption, and ATP production. This circuit may play an important role in repairing neurovascular function after TBI in the peri-injured region by stimulating astrocytic mitochondrial biogenesis.

## 1. Introduction

Korean red ginseng extract (KRGE) has been widely used for healthcare. KRGE and each of its components (ginsenoside) were evaluated for their beneficial effects on the central nervous system (CNS) [1]. Neurovascular cells in the CNS maintain the homeostasis of brain functions. Traumatic brain injury (TBI) results in the disruption of this homeostasis. Recently, regenerative potential of astrocytes among neurovascular cells has been reported in neuroinflammatory diseases [2]. Astrocytes in the CNS play a key role in neuroprotection and energy-metabolic activity in CNS pathological conditions [3].

Heme oxygenase-1 (HO-1) is induced in astrocytes [4,5,6] and exerts various important roles in angiogenesis and mitochondrial biogenesis in them following cerebral ischemia in mice [5,7]. HO metabolites are carbon monoxide (CO), Fe^2+^, and biliverdin [8]. The astrocytic mitochondrial functions are largely associated with upregulation of phospho-AMP-activated protein kinase (p-AMPKα), peroxisome proliferator-activated receptor γ-coactivator-1α (PGC-1α), and estrogen-related receptor α (ERRα) [5,7,9]. Treatment of human astrocytes with CO-releasing molecule-2 for eight hours followed by recovery for 24 h elevates p-AMPKα, PGC-1α, and ERRα proteins [5]. PGC-1α and ERRα play pivotal roles in HO-1/CO-mediated mitochondrial biogenesis and angiogenesis in astrocytes by upregulating cytochrome c and VEGF expression, respectively [5,7].

In TBI, a heterogeneity of astrocytes according to the distance from core injury may exist [2]. In this study, we focused on the peri-injured region in the ipsilateral TBI brain. Since the role of KRGE in astrocytic mitochondrial function in TBI has not been elucidated, we examined the effects of KRGE on astrocytic mitochondrial functions via HO-1 in a brain subjected to TBI (TBI brain). KRGE used for this study contained several ginsenosides, including ginsenoside Rb1 (5.85 mg/g), Rg3s (4.43 mg/g), Rc (2.29 mg/g), Rb2 (2.17 mg/g), Rg3r (2.02 mg/g), Rf (1.37 mg/g), and Rh1 (1.26 mg/g). We examined the signaling cascade associated with AMPKα, PGC-1α, and ERRα expression in HO-1-mediated mitochondrial biogenesis in KRGE-treated astrocytes during the recovery phase of oxygen-glucose deprivation (OGD) injury (OGD/R). KRGE induced HO-1 expression and consequent upregulation of the AMPKα–PGC-1α–ERRα circuit in OGD/R-conditioned astrocytes. In addition, the HO-1–AMPKα–PGC-1α–ERRα signaling pathway contributed to KRGE-mediated mitochondria oxygen consumption and ATP production. In the case of TBI, HO inhibition by Sn(IV) protoporphyrin IX dichloride (SnPP) injection diminished KRGE-mediated mitochondria-related protein expressions, including the expression of p-AMPKα, PGC-1α, ERRα, cytochrome c oxidase subunit 2 (MTCO2), and cytochrome c (Cyt c). Thus, the KRGE-induced HO-1 pathway may contribute to energy metabolism in astrocytes, leading to potential improvement of neurovascular function after TBI.

## 2. Results

### 2.1. KRGE-Induced HO-1 Was Co-Expressed with Astrocytes in the Peri-Injured Region of Traumatic Brain

KRGE was administered to eight-week-old young adult mice via drinking water for three days after TBI (Figure 1A). Since HO-1 plays a key role in angiogenesis and mitochondrial biogenesis [2,10,11], we determined whether KRGE could affect HO-1 expression after TBI. Enhanced HO-1 protein levels were detected in the KRGE-administered mouse brain at approximately bregma −1 to −2 (Figure 1B). Instead, treatment with KRGE (0.015 mg/mL) in the sham group did not increase HO-1 levels (compared to that with no KRGE) (Figure 1B), suggesting that HO-1 expression can be induced by KRGE during brain injury. In the peri-injured region, HO-1 protein expression was observed in the KRGE-administered TBI mice brains, which was co-localized with glial fibrillary acidic protein (GFAP)-stained astrocytes (Figure 1C), suggesting that KRGE regulated HO-1 expression in astrocytes after TBI. Since the HO-1/CO pathway regulates mitochondrial biogenesis and the expression of related proteins in ischemia/reperfusion injury [5], we examined whether KRGE-administered TBI mice brains exhibited high expression of mitochondrial biogenesis-related proteins, such as AMPKα, PGC-1α, ERRα, MTCO2, and cytochrome c (Cyt c). Compared with those of the TBI control (water drinking group), KRGE-administered TBI mice brains showed enhanced expression of proteins, such as p-AMPKα, PGC-1α, ERRα, MTCO2, and Cyt c (Figure 1D), suggesting that KRGE increased mitochondrial functions after TBI.

### 2.2. KRGE Induced the Expression of HO-1 and Mitochondria-Related Proteins in Human Astrocytes

Next, we examined KRGE-induced mitochondrial functions and HO-1 expression in astrocytes in vitro. Following TBI, damaged vascular system may induce hypoxic and glucose-deficient conditions [12]. Thus, human astrocytes were pretreated with OGD for eight hours followed by recovery for 24 h (OGD/R) (Figure 2A). Treatment of human astrocytes with various concentrations of KRGE during recovery increased HO-1 protein expression in a concentration-dependent manner (Figure 2B) without cytotoxicity (Figure 2C). Additionally, the expression of mitochondrial function-related proteins, such as p-AMPKα, PGC-1α, and ERRα, was significantly induced by 250 μg/mL KRGE in OGD/R conditions (Figure 2D,E). The components of mitochondrial electron transport chain (i.e., MTCO2 and Cyt c) were also increased by KRGE, peaking at 250 μg/mL KRGE treatment (Figure 2E). To examine whether KRGE increased functional mitochondrial biogenesis, we determined the mitochondrial membrane potential by staining with Mitotracker. The intensity of Mitotracker was increased by the KRGE treatment in OGD/R conditions compared with that in OGD/R alone conditions (Figure 2F). Therefore, these results suggest that KRGE upregulated astrocytic mitochondrial activity during OGD/R.

### 2.3. KRGE Induced the Expression of Mitochondrial Functional Proteins via HO-1

We examined the role of HO-1 in the energy metabolism of astrocytes exposed to KRGE in OGD/R conditions. Astrocytes subjected to KRGE in OGD/R conditions showed elevated expression of HO-1, p-AMPKα, PGC-1α, and ERRα, and these effects were effectively reduced by HO-1 knockdown using its specific siRNA (si-HO-1) (Figure 3A) or HO inhibitor SnPP (Figure 3B), suggesting that mitochondrial functional regulating proteins are regulated by HO-1 expression and activity.

Treatment of human astrocytes with si-HO-1 significantly reduced the KRGE-induced expression of MTCO2 and Cyt c in OGD/R conditions (Figure 3C). Additionally, oxidative phosphorylation (OXPHOS) complex (I–IV) proteins (i.e., NDUFB8, SDHB, MTCO1, and UQCRC2) and complex V proteins (ATP synthase, ATP5A) were increased by KRGE treatment, and these effects were reduced by HO-1 knockdown (Figure 3D). KRGE induced an increase in Mitotracker intensity (Figure 3E). Hence, astrocyte-derived HO-1 may play a key role in enabling mitochondrial functions, such as OXPHOS and mitochondria membrane potential in response to KRGE treatment in OGD/R conditions.

### 2.4. KRGE Induced AMPKα Activation in Human Astrocytes

Since HO-1 regulated p-AMPKα (T172) in KRGE-treated astrocytes under OGD/R conditions (Figure 3A,B), we examined KRGE-mediated activation of energy-sensing kinase AMPKα in mitochondrial biogenic proteins (i.e., PGC-1α and ERRα) in human astrocytes. AMPK activator 5-aminoimidazole-4-carboxamide-1-β-D-ribonucleoside (AICAR) gradually increased PGC-1α and ERRα protein expression in a concentration-dependent manner (Figure 4A). AICAR-induced increases in PGC-1α and ERRα expression were significantly reduced by the AMPK inhibitor, Compound C (ComC) (Figure 4B). The KRGE-induced increases in p-AMPKα, PGC-1α, and ERRα levels were remarkably reduced by the knockdown of AMPKα under OGD/R conditions (Figure 4C). Moreover, the KRGE-induced increases in MTCO2 and Cyt c levels were remarkably reduced by knockdown of AMPKα (Figure 4D). However, knockdown of AMPKα did not significantly reduce KRGE-induced HO-1 expression (Appendix A). These data collectively suggest that KRGE increases the HO-1–AMPKα axis in astrocytes subjected to OGD/R.

### 2.5. KRGE-Mediated PGC-1α and ERRα Expression Regulated Mitochondrial Functional Proteins

The KRGE-mediated HO-1–AMPKα axis increased the expression of PGC-1α and ERRα in astrocytes subjected to OGD/R (Figure 3 and Figure 4). PGC-1α acts as a master regulator of mitochondrial biogenesis and function via the induction and activation of several nuclear transcription factors, such as ERRα [13]. Additionally, the transcriptional activity of ERRα is largely dependent on the presence of PGC-1α, which is expressed at low basal levels under normal conditions and is induced by energy stress [14]. PGC-1α and ERRα synergistically regulate mitochondrial O_2_ consumption and ATP production [9,15]. We examined whether KRGE-induced PGC-1α and ERRα expression was involved in mitochondrial functions using siRNAs for PGC-1α and ERRα (Figure 5A). ERRα knockdown significantly decreased PGC-1α protein levels and vice versa (Figure 5A), demonstrating the ERRα–PGC-1α circuit. Thus, we tested the relationship between the ERRα–PGC-1α circuit and HO-1 (Appendix A) or p-AMPKα levels (Figure 5A). Knockdown of ERRα significantly decreased the p-AMPKα/AMPKα ratio (Figure 5A) but did not decrease HO-1 expression (Appendix A). Moreover, we could not find synergistic effects of ERRα and PGC-1α on the p-AMPKα/AMPKα ratio compared with the effect of ERRα alone on that ratio (Figure 5A). Knockdown of ERRα markedly reduced the p-AMPKα/AMPKα ratio, leading to the reduction of PGC-1α forming the AMPKα –PGC-1α–ERRα circuit. Marked reduction of MTCO2 in KRGE-treated conditions was observed in the diminished ERRα (Figure 5B). Proteins (i.e., Cyt c, NDUFB8 (complex I), SDHB (complex II), and MTCO1 (complex IV)) related to mitochondrial functions were decreased by knockdown of PGC-1α or ERRα in astrocytes subjected to KRGE treatment under OGD/R conditions (Figure 5B). The KRGE-induced increase in O_2_ consumption and ATP content in OGD/R conditions was reduced upon transfection with si-HO-1, si-AMPKα, si-PGC-1α, or si-ERRα (Figure 5C,D). Collectively, these results suggest that the HO-1–AMPKα–PGC-1α–ERRα pathway may play a key role in KRGE-mediated energy production.

### 2.6. KRGE Induced Mitochondrial Biogenesis via HO-1 Following TBI

Since transient HO-1 induction performs beneficial effects on astrocytes [5,9,16], we examined the in vivo TBI model by injecting HO inhibitor SnPP followed by KRGE administration for three days (Figure 6A). The SnPP treatment diminished the co-expression of HO-1 and GFAP in the peri-injured area (Figure 6B) and decreased HO-1 protein levels without altering the GFAP protein levels (Figure 6C). These data suggest that KRGE upregulates HO-1 expression in astrocytes in an HO activity-dependent manner. Next, mitochondrial biogenesis-related proteins were evaluated. Inhibition of HO activity markedly reduced KRGE-induced protein levels of p-AMPKα, PGC-1α, ERRα, MTCO2, and Cyt c (Figure 6C). In KRGE-administered TBI brains, HO-1 induction and its activity in peri-injured astrocytes may play an essential role in mitochondrial biogenesis by stimulating the sequential p-AMPKα–PGC-1α–ERRα pathway and by upregulation of MTCO2 and Cyt c expression.

## 3. Discussion

KRGE has neuroprotective and therapeutic potential in the CNS [1]. However, the molecular mechanisms underlying KRGE-induced neuroprotection and regeneration after CNS injuries remain unclear. Astrocytes play a key role in maintaining vascular function and neuroprotection in ischemic diseases [2,3,17] by increasing the levels of neurogenic and angiogenic factors [18,19,20] and by stimulating energy production [21]. Here, our novel finding suggests that KRGE may enhance mitochondrial functions in astrocytes by inducing HO-1 expression and activation after TBI.

AMPKα, PGC-1α, and ERRα are associated with the expression of mitochondrial genes [15,22]; however, the association of the AMPKα–PGC-1α–ERRα pathway with KRGE-mediated astrocytic mitochondrial function has not been elucidated, particularly under HO-1-expressing conditions. We found that astrocyte-derived HO-1 exposed to KRGE may play a key role in mitochondrial functions after TBI by inducing AMPKα–PGC-1α–ERRα circuit formation and OXPHOS. The effects of ginsenoside on metabolic diseases in relation to AMPK activation have been investigated in various cell types such as adipocytes, hepatocytes, and myocytes [23]. Moreover, the treatment of cardiomyocytes and neurons with ginsenoside Rc increased the levels of OXPHOS (complex II–IV), ATP production, glucose uptake and mitochondrial pyruvate carrier I/II [24]. Since KRGE used in this study contained various ginsenosides (Rb1, Rg3, Rc, Rb2, Rc, Rh, Rf etc.), we assume that the combination of various ginsenosides may enhance mitochondria functions in astrocytes.

Our data indicate that HO-1 can enhance astrocytic mitochondrial function, as evidenced by increased OXPHOS, mitochondrial oxygen consumption, and energy production, largely through the AMPKα–PGC-1α–ERRα pathway. However, knockdown of AMPKα, PGC-1α, or ERRα did not significantly reduce HO-1 in response to KRGE treatment in astrocytes exposed to OGD/R (Appendix A). Thus, KRGE-mediated HO-1 expression is an upstream signal for AMPKα, PGC-1α, and ERRα. ERRα knockdown significantly suppressed the expression of p-AMPKα (Thr 172, catalytic activity site [25]) and PGC-1α but not HO-1. This result implies that KRGE-mediated ERRα expression can modulate AMPK enzymatic activity as an HO-1 downstream signal. Moreover, PGC-1α knockdown partly suppressed AMPKα activation. Thus, we speculate that ERRα may be a more potent regulator for AMPKα than PGC-1α. Our next study may aim to investigate the mechanisms underlying ERRα-mediated regulation of AMPKα enzymatic activity by checking upstream signaling such as Ca^2+^ influx, liver kinase B1 level, or AMP level. Further studies may also include the mechanisms underlying KRGE-mediated upregulation of HO-1 expression. Additionally, we will examine whether KRGE can facilitate the production of HO metabolites such as CO and bilirubin.

Injury to the somatosensory cortex can induce KRGE-mediated HO-1 expression in the peri-hippocampal region of a TBI brain. The core region of a TBI brain may be a challenging environment for regeneration due to oxygen and glucose deprivation as well as cell death [12]. However, the boundary region of a TBI brain may promote regenerative capacity by enhancing cellular repair mechanisms. We found that KRGE led to HO-1 induction in peri-injured astrocytes, and HO-1 could improve mitochondrial functions through the AMPKα–PGC-1α–ERRα circuit and consequently induce the expression of MTCO2 and Cyt c.

Astrocytes communicate with the neurovascular system [2,26], leading to neurovascular repair after CNS damage [3]. Our previous study demonstrated that a combination of two HO metabolites (i.e., bilirubin and CO) facilitated the AMPKα–PGC-1α–ERRα pathway in astrocytes [5]. KRGE upregulates HO-1 in astrocytes, possibly producing CO and biliverdin. Biliverdin may be converted into bilirubin (antioxidant) through biliverdin reductase [27]. CO also facilitates neurogenesis and improvement of motor and cognitive functions after TBI [12]. Thus, KRGE may stabilize the HO-1-derived CO and bilirubin system, leading to improved astrocytic function in neurovascular units after TBI by activating the AMPKα–PGC-1α–ERRα circuit and mitochondrial functions. Collectively, as a putative therapeutic agent, KRGE may enhance neurovascular regeneration by enhancing the levels and activity of mitochondrial biogenic factors in astrocytes after TBI.

## 4. Materials and Methods

### 4.1. Materials

DMSO (Sigma Aldrich, St. Louis, MO, USA), fetal bovine serum (FBS, Corning, NY, USA), Compound C (ComC, an AMPK inhibitor, Enzo Life Sciences, Farmingdale, NY, USA), AICAR (AMPK activator, Enzo Life Sciences, Farmingdale, NY, USA), and Sn(IV) protoporphyrin IX dichloride (SnPP, HO inhibitor, Frontier Scientific, Logan, UT, USA) were bought. KRGE containing various ginsenosides was obtained from the Korea Ginseng Cooperation (Daejeon, Korea) and stored at 4 °C. Next, a 0.2 g/mL stock solution prepared in filtered distilled water was aliquoted and stored with light protection at −25 °C.

### 4.2. Animals

Male C57BL/6 mice were purchased from Joong Ah Bio Inc. (Suwon, Korea) and maintained under standard conditions with water and food available ad libitum. All mouse experiments were approved by the Animal Ethics Committee of Kangwon National University (approval number KW-181119-2). Additionally, this investigation conformed to the Guide for the Care and Use of Laboratory Animals published by the United States National Institutes of Health. KRGE (0.015 mg/mL) was administered via drinking water for 3 days. The control mice group was administered only water.

### 4.3. Controlled Cortical Impact (CCI) Model for TBI

A CCI model of experimental TBI was established. The 8-week-old male C57Bl/6 mice were deeply anesthetized via 2% inhaled isoflurane in a 7:3 mixture of N_2_O and O_2_ using an isoflurane vaporizer (VetEquip, Livermore, CA, USA), positioned in a stereotaxic apparatus (RWD Life Science, Shenzhen, Guangdong, China). A craniotomy was made at the somatosensory cortex approximately 5 mm over the right hemisphere using a portable drill. Using a CCI device (Leica Impact One; Leica Biosystems, Cat No. 39463923, Buffalo, NY, USA), a 3 mm flat-tip impactor was accelerated down to a 2 mm depth at a velocity of 5 m/s. Injuries were evaluated at 3 days post injury. The animals were assigned randomly to the TBI group. Sham-operated groups only received a craniotomy.

### 4.4. Brain Tissues and Immunohistochemistry

For histological analysis, the mice were anesthetized using isoflurane (1.5%) and N_2_O gas and then transcardially perfused with saline. Using an optimal cutting temperature compound, the freezing brain tissues were sectioned into 20 μm sections by a cryostat (HM525 NX, Thermo Fisher Scientific, Carthage, MO, USA). The sections were incubated with 4% paraformaldehyde for 15 min and washed in phosphate buffered saline (PBS) three times, then incubated with 3% bovine serum albumin for 1 h. The sections were then incubated with mouse anti-HO-1 (1:100, abcam) and rabbit anti-GFAP antibody (1:150, abcam) in PBST (0.1% triton X-100 in PBS) at 4 °C overnight. After washing, the sections were incubated in a mixture of both TRITC-conjugated donkey anti-rabbit IgG (1:200, Jackson ImmunoResearch) and FITC-conjugated donkey anti-mouse IgG (1:100, Jackson ImmunoResearch) for 1 h at room temperature. Between incubations, the tissues were washed with PBST (0.1% Tween-20 in PBS). The sections were visualized using a mounting solution (Fluoro-Gel II with DAPI, Electron Microscopy Sciences, Hatfield, PA, USA). The stained sections were subsequently examined using an inverted phase contrast microscope (Eclipse Ti2-U, Nikon, Minato, Tokyo, Japan).

### 4.5. Cell Culture

Primary human brain astrocytes were acquired from the Applied Cell Biology Research Institute (Kirkland, WA, USA). Astrocytes were cultured in Dulbecco’s Modified Eagle medium (DMEM, HyClone, Omaha, NE, USA) supplemented with 10% FBS. When astrocytes reached 80% density, the media were replaced with serum-free DMEM. The cells were treated with distilled water or KRGE for 24 h in serum-free DMEM.

### 4.6. OGD

After reaching 80% confluency, primary human brain astrocytes were incubated with 0% FBS-containing DMEM media. Hypoxia was induced by perfusing 90% N_2_, 5% CO_2_, and 5% H_2_-containing gas for 15 min in a hypoxia chamber (Billups-Rothenberg, Del Mar, CA, USA), and the chamber was occluded for 8 h at 37 °C in an incubator. After removing the dishes from the chamber, the cells were subjected to 24 h recovery at 37 °C in a normal O_2_ incubator (Thermo Fisher Scientific, Carthage, MO, USA).

### 4.7. Western Blot Analysis

Protein Extraction Solution (RIPA) (Elpis-Biotech, Daejeon, South Korea) was used for cell lysis. Selected amounts of proteins from the cell lysates were combined with a SDS sample buffer (Glycerol 10% (*v*/*v*), Tris-Cl pH 6.8, SDS 2% (*w*/*v*), β-mercaptoethanol 1% (*v*/*v*), and bromophenol blue) and subjected to 100 °C for 5 min. To detect OXPHOS proteins, 15 μg of proteins from the RIPA-mediated cell lysates were combined with a SDS sample buffer and subjected to 37 °C for 5 min. Next, protein samples were separated using SDS-PAGE, and the PVDF membranes (Merck Millipore, Temecular, CA, USA) were blocked in Tris-buffered saline containing 0.1% Tween 20 and 5% skim milk (BD Difco, Burlington, NC, USA). The membranes were incubated with primary antibodies at 4 °C overnight. The primary antibodies used in this study were as follows: AMPKα (1:2000, cell signaling), p-AMPKα (Thr172) (1:2000, cell signaling), Cyt c (1:3000, BD Biosciences, San Jose, CA, USA), HO-1 (1:1000, BD Biosciences, Burlington, NC, USA; 1:1000 Enzo Life Science, Farmingdale, NY, USA; 1:3000 abcam, Cambridge, UK), GFAP (1:3000 BD Biosciences, San Jose, CA, USA), ERRα (1:1000, Novus Biologicals, Centennial, CO, USA), PGC-1α (1:1000, SantaCruz Biotechnology, Dallas, TX, USA), a total OXPHOS antibody cocktail (1:3000, abcam, Cambridge, UK), MTCO2 (1:1000, SantaCruz Biotechnology, Dallas, TX, USA), and β-Actin (1:8000, Sigma Aldrich, Saint Louis, MO, USA). After washing, the membranes were incubated with peroxidase-conjugated secondary antibodies (1:8000, Thermo Fisher Scientific, Carthage, MO, USA) and visualized using enhanced ECL (Elpis-Biotech, Daejeon, South Korea) with appropriate detection equipment (Fusion Solo-Vilber Lourmat, Collegien, France).

### 4.8. Transfection

When astrocytes reached 70% confluency, the cells were transiently transfected with siRNAs for AMPKα, PGC-1α, ERRα, HO-1 (50 nM, SantaCruz Biotechnology, Dallas, TX, USA), or negative control (50 nM, Thermo Fisher Scientific, Carthage, MO, USA) using RNAiMax (Thermo Fisher Scientific, Carthage, MO, USA). After 12 h of recovery, the cells were treated with or without KRGE for 24 h in serum-free DMEM.

### 4.9. Cytotoxicity Assay

The cytotoxicity assay was performed using a lactate dehydrogenase (LDH) assay kit (Roche). Five hundred microliters of total astrocytes medium in 12-well plates was collected and then centrifuged for 7 min at 7000× *g* rpm. Fifty microliters of supernatant was transferred to 96-well plates (CM) in triplicate. Fifty microliters of medium from without cells was used for CM blank. Astrocytes were washed with PBS twice and incubated with 500 µL 5% triton X-100 (Sigma, Saint Louis, MO, USA) in PBS at 37 °C for 20 min. Lysed cells were centrifuged for 5 min at 15,000× *g* rpm. The supernatant (50 µL) was transferred to a 96-well plate (WCL) in triplicate. The supernatant from without cells was used to serve as the WCL blank. A dye solution was mixed with the catalyst at ratio of 45:1. Fifty microliters of the mixed detection LDH assay kit reagent was then added to each of the assay wells on top of the supernatant in rapid succession. The assay plates in 100 µL total volume were then incubated at room temperature with light protection for 20 min, and were then read using a plate reader (BioTek, Winooski, VT, USA) with a reference wavelength of 490 nm. The average values in triplicate were subjected to the following equation: Cytotoxicity (%) = (CM value − CM blank)/((CM value − CM blank) + (WCL value − WCL blank)) × 100

### 4.10. Mitochondrial Activity Assay

Intracellular active mitochondria levels were assessed via quantitative fluorescence imaging using a mitochondria membrane potential-sensitive dye, i.e., Mitotracker (Thermo Fisher Scientific, Waltham, MA, USA). Astrocytes plated on 18 mm round coverslips in 12-well plates were cultured to 80% confluency. The cells were subjected to distilled water or 500 μg/mL KRGE for 23.5 h. Subsequently, the cells were treated with 1 μM of Mitotracker for 30 min. After washing with PBS, fluorescent images of cells were acquired using an inverted phase contrast microscope (Eclipse Ti2-U).

### 4.11. O_2_ Consumption

Live cell O_2_ consumption was assessed using an O_2_ Consumption Rate Assay Kit (Cayman, Ann Arbor, MI, USA). Eighty percent-confluent astrocytes were transfected with the indicated siRNAs and then incubated in a 96-well Black Polystyrene Microplate (Corning, Castle Rock, CO, USA). The cells were exposed to OGD/R with distilled water or 250 μg/mL KRGE for 24 h in serum-free DMEM media, and an O_2_ sensor probe was added to each well. The same procedure with “no cell” was considered as blank. After covering the wells with mineral oil, absorbance was evaluated using a filter combination and emission and excitation wavelengths of 650 and 380 nm, respectively, at 37 °C for 75 min (BioTek, Santa Clara, CA, USA). Absorbance in well with cells minus absorbance in well with “no cell” at 15 min after detecting the O_2_ consumption rate in the normoxic (control) group was set as “1”, and other groups were adjusted to the control group.

### 4.12. Intracellular ATP Assay

The cellular ATP amounts were detected using an ATP colorimetric assay kit (BioVision, Milpitas, CA, USA). Eighty percent-confluent astrocytes were transfected with the indicated siRNAs and exposed to OGD/R with distilled water or 250 μg/mL KRGE in serum-free DMEM media for 24 h and then lysed in 120 μL of ATP assay buffer and centrifuged at 15,000× *g* rpm for 5 min at 4 °C. The collected 50 μL of supernatant was combined with 50 μL of the reaction mixture reagents, and the plates were incubated at room temperature for 30 min while being protected from light, and the absorbance was measured at 570 nm using a colorimetric assay reader (Epoch Microplate Spectrophotometer, BioTek, Santa Clara, CA, USA). The protein content in the lysed cells was quantified using BCA (Thermo Fisher Scientific, Carthage, MO, USA). Then, ATP levels/protein amount (nmol ATP/mg protein) in the control group were set as “100%”, and other groups were adjusted to the control group.

### 4.13. Data Analysis

The intensity of the protein band, which was obtained via Western blot experiments, was evaluated using the ImageJ (http://rsb.info.nih.gov/ij/, accessed on 1 March 2020) program. GraphPad Prism 6 was utilized for overall statistical analysis. Multiple comparisons were evaluated using one-way ANOVA plus Tukey’s test (mean ± SD). Statistical significance was set at *p* < 0.05. * *p* < 0.05; ** *p* < 0.01; *** *p* < 0.001.

## Figures and Tables

**Figure 1 ijms-22-13081-f001:**
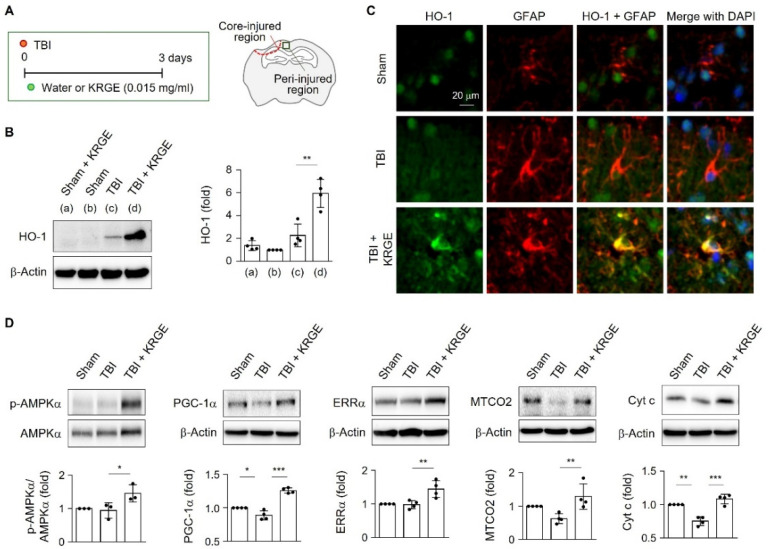
Astrocytic heme oxygenase-1 (HO-1) was expressed in the peri-injured region of a mouse brain with traumatic brain injury (TBI). (**A**) Schematic representation of TBI conditions and Korean red ginseng extract (KRGE) administration (left). Figure for the TBI brain section demonstrates core-injured and peri-injured regions (right). (**B**–**D**) HO-1 protein expression in brain tissues (approximately bregma −1 to −2) was detected in the peri-injured region of the TBI ipsilateral brain. (**B**) HO-1 protein expression was assessed by Western blotting. β-Actin was used as the internal control (*n* = 4). (**C**) Representative image of HO-1 (green) and glial fibrillary acidic protein (GFAP, red) expression in a mouse subjected to TBI (*n* = 3 per group). Scale bar = 20 μm. (**D**) Protein expression in brain sections obtained from sham, TBI, and TBI followed by KRGE (TBI + KRGE) was detected using the indicated antibodies and assessed by Western blotting (*n* = 3 or 4 per group). * *p* < 0.05; ** *p* < 0.01; *** *p* < 0.001.

**Figure 2 ijms-22-13081-f002:**
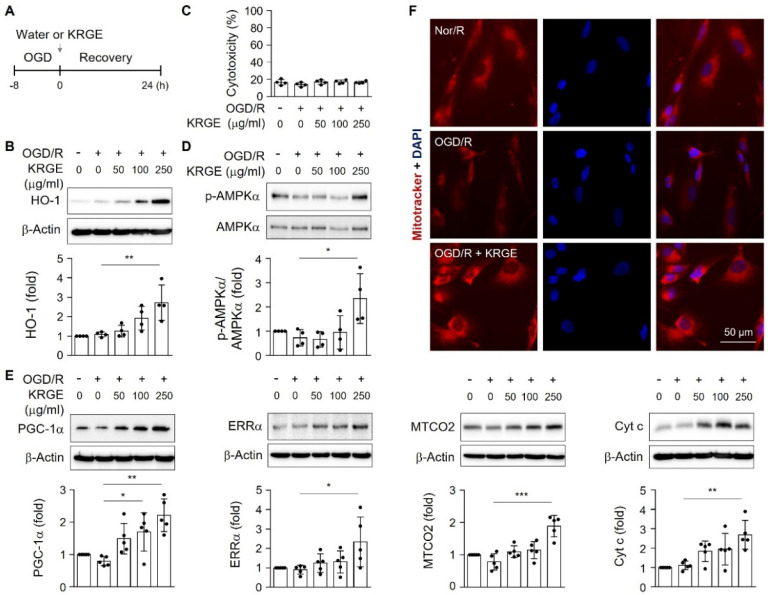
KRGE induces expression of HO-1 and mitochondria-related proteins in human astrocytes. (**A**) Human astrocytes were subjected to oxygen-glucose deprivation (OGD) for 8 h and subsequently treated with KRGE for 24 h. (**B**) Lysed cells obtained from various concentrations of KRGE were collected. HO-1 protein levels were detected by Western blotting (*n* = 4 per group). (**C**) Human astrocytes were subjected to various KRGE concentrations in OGD/R conditions, and cytotoxicity was measured (*n* = 4). (**D**,**E**) Lysed cells obtained from various concentrations of KRGE were collected. Indicated protein levels were detected by Western blotting (*n* = 4 or 5 per group). * *p* < 0.05; ** *p* < 0.01; *** *p* < 0.001. (**F**) Representative image of the Mitotracker (red) and DAPI (blue) staining in human astrocytes (*n* = 5 per group). Scale bar = 50 μm.

**Figure 3 ijms-22-13081-f003:**
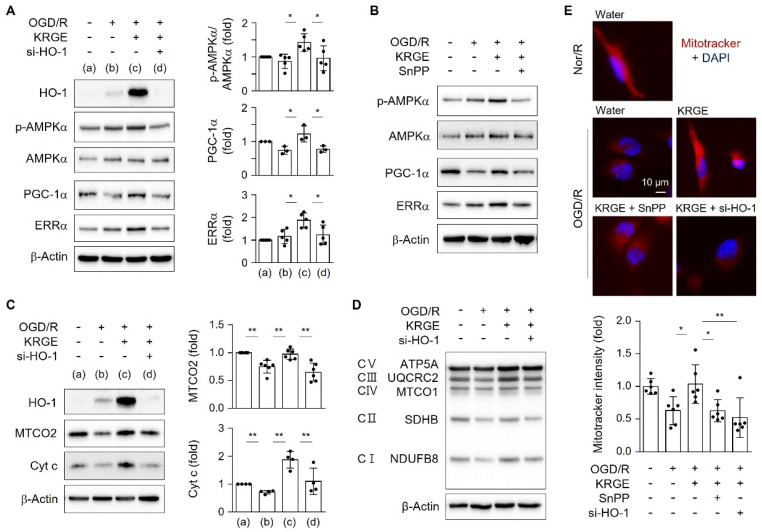
KRGE-mediated HO-1 expression enhanced mitochondrial functional regulating proteins. (**A**) Human astrocytes were transfected with 50 nM siRNAs as the control or HO-1 (si-HO-1) and subjected to OGD/R followed by treatment with 250 μg/mL KRGE. Target protein levels in lysed cells were detected via Western blotting. (**B**) Astrocytes were subjected to OGD/R and co-treated with 25 μM Sn(IV) protoporphyrin IX dichloride (SnPP) and 250 μg/mL KRGE for 24 h. Target protein levels in lysed cells were detected via Western blotting (*n* = 3). (**C**) Under same (**A**) conditions, cytochrome c oxidase subunit 2 (MTCO2) (*n* = 6) and Cyt c (*n* = 4) were detected via Western blotting. (**D**) Oxidative phosphorylation (OXPHOS) complexes (C I–IV) and ATP synthase (ATP5A) (C V) were demonstrated via Western blotting (*n* = 3). (**E**) Representative image of the Mitotracker (red) and DAPI (blue) staining in human astrocytes (*n* = 6 per group). Scale bar = 10 μm. * *p* < 0.05; ** *p* < 0.01.

**Figure 4 ijms-22-13081-f004:**
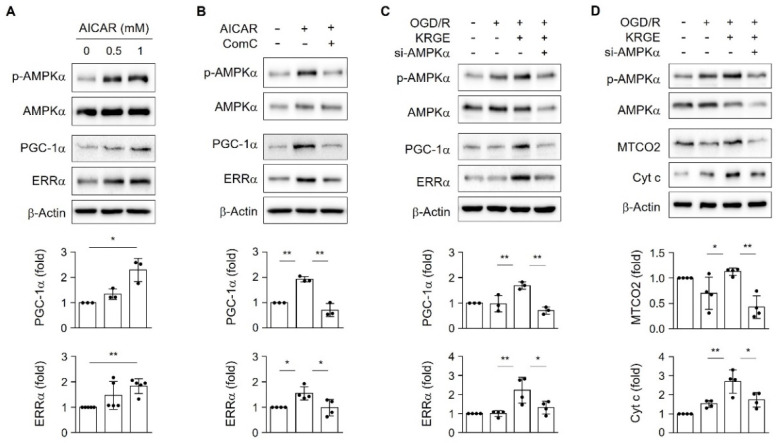
AMP-activated protein kinase α (AMPKα) regulated KRGE-mediated peroxisome proliferator-activating receptor-γ coactivator-1α (PGC-1α) and estrogen-related receptor α (ERRα) expression. (**A**) When astrocytes reached 90% confluency, the culture media were replaced with serum-free Dulbecco’s Modified Eagle Medium (DMEM). After 1 h, the cells were treated with 0, 0.5, or 1 mM 5-aminoimidazole-4-carboxamide-1-β-D-ribonucleoside (AICAR) for 4 h. The expression of indicated proteins was detected via Western blotting. (**B**) When astrocytes reached 90% confluency, culture media were replaced with serum-free DMEM. After 1 h, the cells were pretreated with 10 μM Compound C (ComC) for 15 min and treated with 1 mM AICAR for 4 h. The expression of indicated proteins was detected via Western blotting. (**C**,**D**) Astrocytes were transfected with 50 nM control or 50 nM AMPKα siRNA (si-AMPKα) and subjected to OGD/R followed by treatment with 250 μg/mL KRGE. Indicated protein levels were analyzed using Western blotting and quantified. (**A**–**D**) *n* = 3–5 independent experiments. * *p* < 0.05; ** *p* < 0.01.

**Figure 5 ijms-22-13081-f005:**
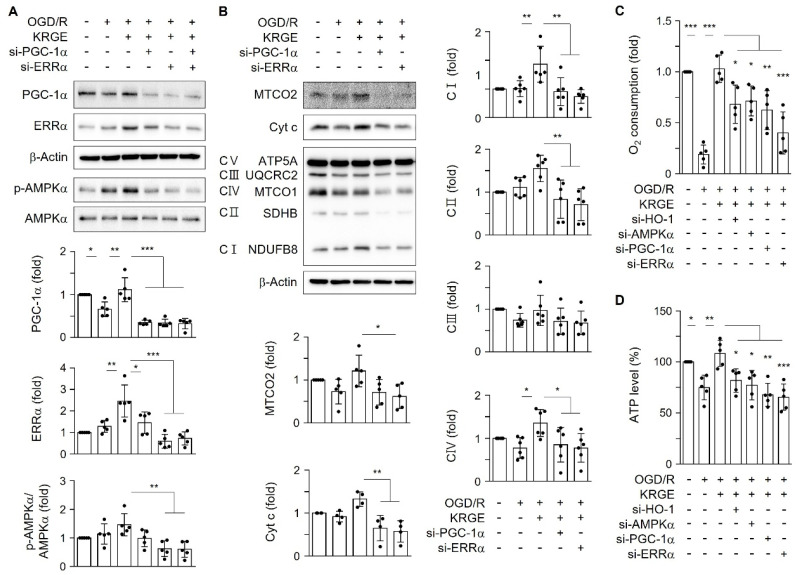
The KRGE-mediated PGC-1α-ERRα axis regulated mitochondrial functional proteins. (**A**,**B**) Astrocytes were transfected with 50 nM siRNAs for PGC-1α (si-PGC-1α) or/and ERRα (si-ERRα). Subsequently, the cells were subjected to OGD/R followed by treatment with 250 μg/mL KRGE. Indicated protein levels were detected by Western blotting. (**C**) Mitochondrial oxygen consumption was detected (*n* = 5). (**D**) ATP levels are shown (*n* = 5). * *p* < 0.05; ** *p* < 0.01; *** *p* < 0.001.

**Figure 6 ijms-22-13081-f006:**
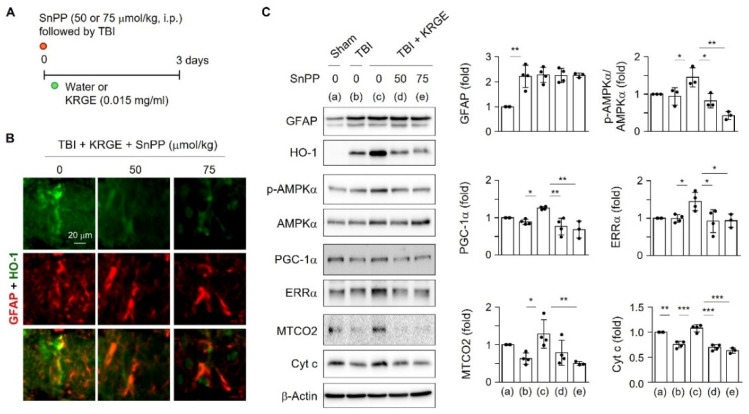
HO inhibition suppressed mitochondria-regulating proteins. (**A**) Schematic representation of SnPP, TBI, and KRGE conditions. (**B**) Representative image of HO-1 (green) and GFAP (red) co-expression in a mouse brain subjected to TBI, followed by KRGE treatment with or without SnPP (*n* = 3 per group). Scale bar = 20 μm. (**C**) The brain sections (approximately bregma −1 to −2) obtained after TBI followed by KRGE treatment with or without SnPP were detected with the indicated antibodies and assessed by Western blotting (*n* = 3 or 4 per group). * *p* < 0.05; ** *p* < 0.01; *** *p* < 0.001.

## Data Availability

The data presented in this study are contained within the article. Original data are available on request from the corresponding author.

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
