# Peer review of "Korean Red Ginseng Improves Astrocytic Mitochondrial Function by Upregulating HO-1-Mediated AMPKα–PGC-1α–ERRα Circuit after Traumatic Brain Injury"

_ijms, 2021, doi:10.3390/ijms222313081_

Round 1

Reviewer 1 Report

Reviewer’s comments on the article ,, Korean red ginseng improves astrocytic mitochondrial biogenesis by upregulating HO-1-mediated AMPKα‒PGC-1α‒ERRα  circuit after traumatic brain injury” by Minsu Kim, Joohwan Kim, Sunhong Moon, Bo Young Choi, Sueun Kim, Hui Su Jeon, Sang Won Suh, Young-Myeong Kim and Yoon Kyung Choi

The paper presents the results of experimental studies of influence of Korean red ginseng on energy metabolism in astrocytes. It needs to be admitted that large work has been put to obtain presented results in the field of molecular-based experiments. Whereas the manuscript is interesting for the readers it has the one main weakness – there is no  information about the type of Korean  ginseng, especially its chemical composition. Therefore I strongly suggest to present characterization of phytocompounds present in studied extract. As a result in the discussion section there is no correlation about ginseng composition and its influence on, i.e. AMPK activation or other activity. I have also some other comments:

  • please, explain why 250 µg/mL concentration of ginseng extract was used for in vitro studies; have the Authors checked higher concentrations?
  • in the AMPK-involvement study the Authors used aicar as AMPK activator or siAMPKa; have the Authors checked other kinases which are responsible for AMPK activation, i.e. LKB1;
  • in the Introduction there is not sufficient explanation about the aim of the study;
  • it is strongly recommended to present larger cell’s photos – this will significantly enrich the manuscript;
  • In figures with bar graphs it is recommended to present data as a medium value +/- SD;
  • Figure 3D and Figure 5B, please describe with details the method to obtain the presented blots.
  • do the Authors have any recommendation of using of KGE in mitochondrial biogenesis improvement, especially in regard to its bioavailability.

Since several issues should be explained and corrected, therefore major revision is recommended.

Author Response

Comment #1. please, explain why 250 µg/mL concentration of ginseng extract was used for in vitro studies; have the Authors checked higher concentrations?

Response #1. We used various concentration of ginseng extract, and found that higher concentrations of ginseng (i.e., 500 µg/mL) more than 250 µg/mL have smaller effects on protein levels of p-AMPKa, MTCO2 and cytochrome c under KRGE-treated OGD/R conditions. Thus, we decided to use 250 µg/mL KRGE for further study.

Comment #2. in the AMPK-involvement study the Authors used aicar as AMPK activator or siAMPKa; have the Authors checked other kinases which are responsible for AMPK activation, i.e. LKB1;

Response #2. We did not check other kinases for AMPKa activation yet. We will check them in the near future.

Comment #3. in the Introduction there is not sufficient explanation about the aim of the study;

Response #3. We have added explanation and indicated these in red font in our manuscript.

Comment #4. it is strongly recommended to present larger cell’s photos – this will significantly enrich the manuscript;

Response #4. We presented larger cell’s photos in Figure 2D and Figure 3E.

Comment #5. In figures with bar graphs it is recommended to present data as a medium value +/- SD;

Response #5. We used ‘mean with SD’ in GraphPad Prism software when we showed bar graphs.

Comment #6. Figure 3D and Figure 5B, please describe with details the method to obtain the presented blots.

Response #6. We have added details in Method section and have indicated these in red font in our manuscript.

Comment #7. do the Authors have any recommendation of using of KGE in mitochondrial biogenesis improvement, especially in regard to its bioavailability.

Response #7. As the other reviewers’ comments, we tried to reveal more precise mechanisms of mitochondrial metabolism. Thus, we checked the ATP content and mitochondrial O2 consumption and added these data in Figure 5C-D. In addition, we toned down our results by replacing ‘biogenesis’ with ‘function’.

In TBI brain, we will detect astrocytes bioavailability regulated by KRGE in the near future.

Reviewer 2 Report

In this study Kim et al. found that Korean red ginseng can improve astrocytic mitochondria biogenesis. This effect may play an important role in repairing neurovascular function after traumatic brain injury.

There are several limitations of this study which should be mentioned:

  • mitochondria biogenesis is different from mitochondrial function
  • protein expression levels do not always coincide with an increase in protein function or enzyme activity

Therefore, suggested mechanisms for the observed effects are poorly supported by the reported data. For example, Authors write “Our data indicate that HO-1 can improve oxygen-dependent energy metabolism through mitochondrial biogenesis largely through the AMPKα-PGC-1α-ERRα pathway.

How Authors investigate mitochondrial metabolism?

MitoTracker stains mitochondria in live cells and its accumulation is dependent upon membrane potential, which is an indicator of mitochondrial activity. To investigate “oxygen-dependent energy metabolism”, Authors should carry out Seahorse assays to obtain OCR (oxygen consumption rates) values or polarographic assay to measure oxygen consumption or some specific enzyme activities.

Moreover, the manuscript often lacks clarity and the scientific question proposed in the manuscript should be clearly presented.

In my opinion the impact of the research is too low to be published in IJMS.

Author Response

Comment #1. There are several limitations of this study which should be mentioned:

  • mitochondria biogenesis is different from mitochondrial function
  • protein expression levels do not always coincide with an increase in protein function or enzyme activity

Therefore, suggested mechanisms for the observed effects are poorly supported by the reported data. For example, Authors write “Our data indicate that HO-1 can improve oxygen-dependent energy metabolism through mitochondrial biogenesis largely through the AMPKα-PGC-1α-ERRα pathway.

Response #1: We have thought that KRGE may improve mitochondrial biogenesis in astrocytes because PGC-1a has been known as a major regulator of mitochondrial biogenesis. However, we also agree with your opinion. Therefore, we changed our title to “Korean red ginseng improves astrocytic mitochondrial function by upregulating HO-1-mediated AMPKα‒PGC-1α‒ERRα circuit after traumatic brain injury”.

Comment #2. How Authors investigate mitochondrial metabolism?

Response #2: To reveal more precise mechanisms of mitochondrial metabolism, we checked the ATP content and mitochondrial O2 consumption and added these data in Figure 5C-D. In addition, we toned down our results by replacing ‘metabolism’ with ‘function’.

Comment #3: MitoTracker stains mitochondria in live cells and its accumulation is dependent upon membrane potential, which is an indicator of mitochondrial activity. To investigate “oxygen-dependent energy metabolism”, Authors should carry out Seahorse assays to obtain OCR (oxygen consumption rates) values or polarographic assay to measure oxygen consumption or some specific enzyme activities.

Response #3: We are sorry for the misinterpretation regarding MitoTracker stains. We evaluated MitoTracker data as membrane potential. In addition, we investigated OCR values and added them in Figure 5C.

Comment #4: Moreover, the manuscript often lacks clarity and the scientific question proposed in the manuscript should be clearly presented.

Response #4: We tried to improve clarity in the manuscript by performing experiments (i.e., evaluating mitochondrial oxygen consumption and ATP content) and by replacing ‘biogenesis’ with ‘function’. We have indicated these in red font in our manuscript.

Reviewer 3 Report

Kim et al. proposed the manuscript « Korean red ginseng improves astrocytic mitochondrial biogenesis by upregulating HO-1-mediated AMPKa-PGC-1a-ERRa circuit after traumatic brain injury ».

The article is well-written. The results are interesting. Although, results need to be improved. The article requests a major revision to better understand the molecular mechanism behind KRGE in TBI condition.

Here are the comments :

(1) Figure 1, in vivo : The authors have to explain why they used 0.015 mg/ml KRGE. Did the author try other KRGE concentration ?

(2) Figure 1, in vivo :  An important group is missing in the study : KRGE alone without TBI. The authors have to show the effects of KRGE alone considering the circuit molecular mechanism discussed here, and also considering HO-1 and GFAP levels.

(3) Figure 2, in vitro : the group of KRGE alone has to be included in the study, including with different KRGE concentrations.

(4) Figure 2, in vitro : Mitotracker red dye is sensitive to mitochondrial membrane potential. So the conclusion about mitochondrial mass is not true. Only Mitotracker Deep Red is insensitive to MMP. The author have to change their conclusion and add mitotracker deep red if they want to reach mitochondrial mass. Otherwise, authors have to estimate the mitochondrial membrane potential with specific dye.

(5) Figure 3, in vitro : same comment than (4) for mitotracker red wrong interpretation. Mitochondrial membrane potential has to be considered.

(6) Figure 3, in vitro : talking about bioenergetic metabolism, the authors did not estimate the ATP content, from mitochondria, and from glycolysis. This is a crucial point to consider.

(7) The authors have to remove all « data not shown ». The authors are not allowed to use any « data not shown » to argue or to make a conclusion. For example, line 162, line 246.

(8) Figure 4 : What is the rational putting AICAR, ComC and KRGE in the same study ?

(9) For all croped western blot, the authors have to provide in supplementary materials the correspoding full membrane without cropping.

All the points have to be investigated to further improve the study and to consolidate the conclusion.

Author Response

Comment #1: Figure 1, in vivo : The authors have to explain why they used 0.015 mg/ml KRGE. Did the author try other KRGE concentration ?

Response #1: For in vivo assay, we referred to a paper by Adam et al [1]. The authors administered KRGE (1.32 mg/2 ml per day) using oral gavage to rats weighing 220~250 g. As a single 8-week-old mouse consumes approximately 10 ml water per day, we set the KRGE concentration (0.015 mg/ml) in mice weighing 26 g, i.e., 0.15 mg/10 ml per day for one mouse.

We did not try other KRGE concentration after TBI yet; however, we will do that experiment in the near future.

References

  1. Adam, G. O.; Kim, G. B.; Lee, S. J.; Lee, H.; Kang, H. S.; Kim, S. J., Red Ginseng Reduces Inflammatory Response via Suppression MAPK/P38 Signaling and p65 Nuclear Proteins Translocation in Rats and Raw 264.7 Macrophage. Am J Chin Med 2019, 47, (7), 1589-1609.

Comment #2: Figure 1, in vivo :  An important group is missing in the study : KRGE alone without TBI. The authors have to show the effects of KRGE alone considering the circuit molecular mechanism discussed here, and also considering HO-1 and GFAP levels.

Response #2: We performed experiments using KRGE (0.015 mg/ml) alone, i.e., without TBI and detected the HO-1 protein levels. We added these data in Figure 1B. We could not find any significant increases in HO-1 in the sham group upon using KRGE alone. Instead, KRGE treatment in conjunction with TBI markedly increased the HO-1 levels in the peri-injured brain regions. Therefore, based on different HO-1 expression levels (with or without brain injury), we assume that KRGE may play a key role in HO-1-mediated neurovascular repair probably in astrocytes during the recovery phase after TBI.

Comment #3: Figure 2, in vitro : the group of KRGE alone has to be included in the study, including with different KRGE concentrations.

Response #3: In vivo data in Figure 1B demonstrates that KRGE (0.015 mg/ml) alone without TBI does not significantly induce the expression of the HO-1 protein. Same amount of KRGE in conjunction with TBI markedly increased the expression of the HO-1 protein. Based on these data, KRGE-induced HO-1 may play different roles in pathophysiological conditions. We would like to investigate the precise mechanisms of action and role of KRGE in physiologic conditions later.

Comment #4: Figure 2, in vitro : Mitotracker red dye is sensitive to mitochondrial membrane potential. So the conclusion about mitochondrial mass is not true. Only Mitotracker Deep Red is insensitive to MMP. The author have to change their conclusion and add mitotracker deep red if they want to reach mitochondrial mass. Otherwise, authors have to estimate the mitochondrial membrane potential with specific dye.

Response #4: We apologize for our wrong interpretation. We changed MitoTracker red stains as mitochondrial membrane potential. The changed part has been depicted in red font in our revised manuscript.

Comment #5: Figure 3, in vitro : same comment than (4) for mitotracker red wrong interpretation. Mitochondrial membrane potential has to be considered.

Response #5: Per your comment, we changed MitoTracker red stains as mitochondrial membrane potential. In addition, we performed more experiments (i.e., evaluation of ATP content assay and O2 consumption) for evaluating the mitochondrial functions and added data in Figure 5C-D.

Comment #6: Figure 3, in vitro : talking about bioenergetic metabolism, the authors did not estimate the ATP content, from mitochondria, and from glycolysis. This is a crucial point to consider.

Response #6: We estimated the ATP content and added that data in Figure 5C-D.

Comment #7: The authors have to remove all « data not shown ». The authors are not allowed to use any « data not shown » to argue or to make a conclusion. For example, line 162, line 246.

Response #7: We added siAMPKa-mediated HO-1 expression data in Supplementary Figure 1A. Thus, we removed ‘data not shown’ and demonstrated the data in Supplementary Figure 1A.

Comment #8: Figure 4 : What is the rational putting AICAR, ComC and KRGE in the same study ?

Response #8: We showed that in the graphical figure. We used AICAR for AMPK activation and found that AICAR (AMPK activator) upregulates the levels of p-AMPKa, PGC-1a and ERRa. To confirm it, we also treated the cells with ComC (AMPK inhibitor). The below figure may explain it.

Comment #9: For all croped western blot, the authors have to provide in supplementary materials the correspoding full membrane without cropping.

Response #9: We already uploaded full membrane data when we submitted this manuscript in IJMS. The updated full membrane data have been presented below.

Round 2

Reviewer 1 Report

The most important thing is that the Authors have not answered to my concerns and the manuscript still requires the major revision, since it is not suitable for publication in present form. First of all, there is still no data in the manuscript about the phytocompounds present in Korean ginseng, i.e. the identification of its components, as well as the quantitation of identified phytocompounds. The Author even did not try to explain the lack of studies in this regard – and it is meaningless and without scientific background to present biological activity of extract with unknown composition. The second issue is that the Authors have not checked the cytotoxic activity in the in vitro study, therefore answer suggesting that 500 µg/mL had smaller effect on p-AMPK, MTCO2 and cytochrome c under KRGE-treated OGD/R conditions than 250 µg/mL, does not support the selection of the 250 µg/mL concentration (maybe it was cytotoxic?). Thirdly, the Authors did not discuss (in Discussion section) the other mechanisms which may lead to activation of AMPK – the chemical analysis of extract contents may be suitable to find information about biological activity of the identified constituents, as well as can allow to explain observed results and match them with other molecular studies based on searching of phytocompounds activity. And the last, but not the least, I asked the Authors about the detailed description of obtaining blots presented in Figure 3D and Figure 5B – unfortunately, I am not satisfied by the Author’s answer; therefore, again, please, explain with technical details the methodology for presenting these blots, which strongly differ from the others presented within the manuscript –were the membranes incubated with mixture of primary antibodies or a single antibody? were the membranes stripped? why in some cases separated blots are presented e.g. Figure 5A, Figure 3 A, B, C; and what was the rationale for choosing particular methodology in this regard?

Author Response

Reviewer #1

The most important thing is that the Authors have not answered to my concerns and the manuscript still requires the major revision, since it is not suitable for publication in present form.

  1. First of all, there is still no data in the manuscript about the phytocompounds present in Korean ginseng, i.e. the identification of its components, as well as the quantitation of identified phytocompounds. The Author even did not try to explain the lack of studies in this regard – and it is meaningless and without scientific background to present biological activity of extract with unknown composition.

: We received KRGE from Korea Ginseng Cooperation with evaluation of each major ginsenoside. Per your valuable comment, we demonstrated KRGE information containing major ginsenosides (i.e., quantification). We added this information in Line 57-59 in this manuscript.

Line 57-59: KRGE used for this study contains several ginsenosides including ginsenoside Rb1 (5.85 mg/g), Rg3s (4.43 mg/g), Rc (2.29 mg/g), Rb2 (2.17 mg/g), Rg3r (2.02 mg/g), Rf (1.37 mg/g) and Rh1 (1.26 mg/g).

  1. The second issue is that the Authors have not checked the cytotoxic activity in the in vitro study, therefore answer suggesting that 500 µg/mL had smaller effect on p-AMPKa, MTCO2 and cytochrome c under KRGE-treated OGD/R conditions than 250 µg/mL, does not support the selection of the 250 µg/mL concentration (maybe it was cytotoxic?).

: In Figure 1A (below data), we could not find any cytotoxicity from various concentrations of KRGE in OGD/R conditions. In addition, we found that 500 mg/mL KRGE might have smaller effects on various mitochondria-related proteins (i.e., ERRa, cytochrome c)’ levels (Figure 1B). Therefore, we selected 250 mg/mL KRGE for further experiments.

In this manuscript, we added cytotoxicity data (KRGE 0 – 250 mg/mL) in Figure 2B.

Figure 1. (A) Cytotoxicity assay. (B) Western blotting.

  1. Thirdly, the Authors did not discuss (in Discussion section) the other mechanisms which may lead to activation of AMPK – the chemical analysis of extract contents may be suitable to find information about biological activity of the identified constituents, as well as can allow to explain observed results and match them with other molecular studies based on searching of phytocompounds activity.

: There are several ginsenosides showing AMPK activation in various cell types. In human astrocytes, we could not find any specific ginsenoside(s) to promote mitochondrial functions through HO-1-AMPKa pathway. Therefore, we assume that combination of various ginsenosides may enhance mitochondria functions in astrocytes.

We added this information in Line 254-260 in this manuscript.

[Line 254-260] The effects of ginsenoside on metabolic diseases in relation to AMPK activation have been investigated in various cell types such as adipocytes, hepatocytes and myocytes [23]. Moreover, treatment of cardiomyocytes and neurons with ginsenoside Rc increased the levels of OXPHOS (complex II-IV), ATP production, glucose uptake and mitochondrial pyruvate carrier I/II [24]. Since KRGE used in this study contains various ginsenosides (Rb1, Rg3, Rc, Rb2, Rc, Rh, Rf etc.), we assume that combination of various ginsenosides may enhance mitochondria functions in astrocytes.

  1. And the last, but not the least, I asked the Authors about the detailed description of obtaining blots presented in Figure 3D and Figure 5B – unfortunately, I am not satisfied by the Author’s answer; therefore, again, please, explain with technical details the methodology for presenting these blots, which strongly differ from the others presented within the manuscript –were the membranes incubated with mixture of primary antibodies or a single antibody? were the membranes stripped? why in some cases separated blots are presented e.g. Figure 5A, Figure 3 A, B, C; and what was the rationale for choosing particular methodology in this regard?

: We used OXPHOS antibody containing five antibodies. Below is the antibody information.

Other western blot bands are obtained from each single antibody.

Reviewer 2 Report

The Authors have addressed all my concerns and I have no further comments. 

As far as I am concerned, the manuscript is now acceptable to be published.

Author Response

Thank you for your comments.

Reviewer 3 Report

I do not have further comments.

Author Response

Thank you for your comments.

Round 3

Reviewer 1 Report

In the response the Authors have given more detailed explanation. Still, the biggest concern is connected with the lack of chemical characterization of the studied extract; the information given is residual; the presentation of the content of some ginsenosides shows rather the lack of information about other components of extracts, for example the ginsenosides content Rb1 (5.85 mg/g), Rg3s (4.43 mg/g), Rc (2.29 mg/g), Rb2 (2.17 mg/g), 58 Rg3r (2.02 mg/g), Rf (1.37 mg/g) and Rh1 (1.26 mg/g) gives the sum equal to 19.39 mg/g – therefore the question is what are the other components of extract? The Authors even did not explain what “g” means  - is it a dry mass of the plant or extract obtained from the plant, etc. The lack of detailed characterization of studied extract does not give the meaning to present biological activity of this unknown extract. What is more, there are many scientific articles explaining the influence of ginsenosides (potentially also phenolic compounds) on AMPK or even heme oxygenase 1, therefore the Authors should put more effort on presentation of identified compounds and their biological effect, and discuss it.

My additional comment is connected to the presented lack of cytotoxic activity of KRGE – the Authors did not explain the method used for cytotoxicity evaluation; also presented data demonstrates that even in control cells cytotoxicity is equal to circa 15%, therefore it is very high and comparable with cells incubated with extract. This is quite surprising.

Since there was a huge effort taken for biological studies I strongly suggest to perform additional analysis of extract composition and submit the manuscript again. Without this crucial data, in my opinion, the manuscript should not be published in International Journal of Molecular Sciences, which is a journal with very high scientific impact.